# Efficacy of Topical Essential Oils in Musculoskeletal Disorders: Systematic Review and Meta-Analysis of Randomized Controlled Trials

**DOI:** 10.3390/ph16020144

**Published:** 2023-01-19

**Authors:** Eszter Bakó, Péter Fehérvári, András Garami, Fanni Dembrovszky, Emese Eszter Gunther, Péter Hegyi, Dezső Csupor, Andrea Böszörményi

**Affiliations:** 1Department of Pharmacognosy, Semmelweis University, 26 Üllői Str., H-1089 Budapest, Hungary; 2Centre for Translational Medicine, Semmelweis University, 26 Üllői Str., H-1089 Budapest, Hungary; 3Department of Biostatistics, University of Veterinary Medicine, 2 István Str., H-1078 Budapest, Hungary; 4Department of Thermophysiology, Institute for Translational Medicine, Medical School, University of Pécs, 12 Szigeti Str., H-7624 Pécs, Hungary; 5Institute for Translational Medicine, Medical School, University of Pécs, 12 Szigeti Str., H-7624 Pécs, Hungary; 6Petz Aladár County Teaching Hospital, Vasvári Str. 2-4, H-9024 Győr, Hungary; 7Division of Pancreatic Diseases, Heart and Vascular Center, Semmelweis University, 23-26 Baross Str., H-1085 Budapest, Hungary; 8Institute of Clinical Pharmacy, Faculty of Pharmacy, University of Szeged, 8 Szikra Str., H-6725 Szeged, Hungary

**Keywords:** phytotherapy, pain, aromatherapy, massage, arthritis, low back pain, dermal application, essential oil

## Abstract

Essential oils (EOs) are widely used topically in musculoskeletal disorders (MSDs); however, their clinical efficacy is controversial. Our aim was to find evidence that topical EOs are beneficial as an add-on treatment in MSDs. We performed a systematic review and meta-analysis to summarize the evidence on the available data of randomized controlled trials (RCTs). The protocol of this work was registered on PROSPERO. We used Web of Science, EMBASE, PubMed, Central Cochrane Library and Scopus electronic databases for systematic search. Eight RCTs were included in the quantitative analysis. In conclusion, EO therapy had a favorable effect on pain intensity (primary outcome) compared to placebo. The greatest pain-relieving effect of EO therapy was calculated immediately after the intervention (MD of pain intensity = −0.87; *p* = 0.014). EO therapy had a slightly better analgesic effect than placebo one week after the intervention (MD of pain intensity = −0.58; *p* = 0.077) and at the four-week follow-up as well (MD of pain intensity = −0.52; *p* = 0.049). EO therapy had a beneficial effect on stiffness (a secondary outcome) compared to the no intervention group (MD = −0.77; *p* = 0.061). This systematic review and meta-analysis showed that topical EOs are beneficial as an add-on treatment in reducing pain and stiffness in the investigated MSDs.

## 1. Introduction

Musculoskeletal disorders (MSDs) are major public health issues all over the world because they cause long-term pain and physical disabilities and reduce people’s ability to work [1]. MSDs cover the problems related to the different areas of the body, i.e., the back, the neck, the shoulder, and the limbs can be affected, and even joints or tissues. The main purpose of the treatment is to relieve pain and ameliorate stiffness and other physical conditions.

Pharmacological treatments of MSDs include topical or oral analgesics (non-steroidal anti-inflammatory drugs (NSAIDs), paracetamol, tramadol, and opioids), chondroitin sulphate, glucocorticoids, disease-modifying antirheumatic drugs (DMARDs) and other drugs [2]. Painkillers may have serious side effects, especially in the case of long-term usage (in chronic disorders) and in the case of high doses [3]. A well-chosen, evidence-based phytotherapy could be beneficial in pain treatment because it can reduce the amount of the necessary analgesic medicines or prolong the length of the effective treatment before the loss of efficacy in pain management [4].

Essential oils (EOs) are complex secondary metabolites that are produced by aromatic plants; they are composed of many apolar or semi-polar volatile constituents with low molecular mass. The largest group of them are terpenoids, but phenolic compounds are also dominant. An EO generally consists of 10–50 compounds with various structures, which can be carbohydrates or oxygen-containing compounds such as alcohols, ketones, aldehydes, ethers and esters [5,6]. The chemical composition determines the biological activity of an EO [7], but even one single component can be highly bioactive (e.g., menthol or camphor). EOs are mostly inhaled or applied topically (beside some less significant uses like oral, vaginal or rectal application); a common method is via massage of the chosen oil. The smell and even the touch are important for the parasympathetic effect that facilitates relaxation and, consequently, causes decrease in pain intensity [8]. Furthermore, EO constituents act on different transient receptor potential channels (TRP channels) which have important roles in pain, heat and cold sensation [9]. Lavender, peppermint, rosemary, eucalypt and chamomile EOs are used to treat MSD traditionally [10,11,12,13,14]. The purpose of their usage is to decrease musculoskeletal pain and inflammation and to improve the blood circulation. They also have a cooling and local anaesthetic effect as well as a muscle relaxation effect, and alleviate depression associated with long-term pain [8]. The pain-relieving effect of different EOs has been confirmed in several animal experiments, where EOs have usually been applied orally or intraperitoneally [15,16,17,18,19]. Despite their popularity and long-standing traditional use, there is little evidence on the clinical efficacy of topically applied EOs.

The purpose of this systematic review and meta-analysis is to evaluate the efficacy of topically used EOs and to assess the hypothesis that topical EO therapy is beneficial as an add-on treatment in MSDs. Furthermore, based on our results, our aim is to provide evidence-based recommendations for healthcare professionals.

## 2. Results

### 2.1. Search and Selection

With the searching process, 752 articles were collected. Duplication removal resulted in 518 records for the subsequent title and abstract selection phase. After full text screening, altogether 12 studies [20,21,22,23,24,25,26,27,28,29,30,31] were included in the systematic review. More details on the search and selection process are presented in the PRISMA (Preferred Reporting Items for Systematic Reviews and Meta-Analyses) flowchart in Figure 1.

### 2.2. Basic Characteristics of Included Studies

Baseline characteristics of the randomized controlled trials (RCTs) included in this paper are presented in Table 1. The trials were carried out between 2004 and 2020. All patients (817) included in the studies had MSD. Three trials were carried out in Iran, four in China, and one in Turkey, Taiwan, USA, and Egypt, respectively.

### 2.3. Qualitative Synthesis of Results

In the EO therapy group, the EOs were applied topically as a complementary treatment in addition to the conventional therapy of MSDs. In the Placebo group, a placebo product (a vegetable carrier oil or an ointment without any EOs) was used as an add-on treatment in MSDs. In the No intervention group, patients did not receive either EO therapy or other additional interventions, only the conventional therapy. The EO-containing products and placebo products were applied by massage in most of the trials. The length of the interventions differed in the trials; they were mostly three or four weeks long.

Different EOs were used in the trials. Lavender EO was used in seven trials, and the applied concentrations were between 1.5% and 3%. In the case of other EOs, the applied concentrations were between 0.5% and 2.5%. In one case [24], an ointment contained 20% of EO. More details of EOs applied in the trials are presented in Table A1 in the Appendix B.

All investigated trials concluded that EO therapy might be a beneficial treatment for pain intensity (primary outcomes). The following conditions were investigated in the trials: knee osteoarthritis (OA) and hand OA, rheumatoid arthritis, low back pain, carpal tunnel syndrome (CTS) and neck pain. Due to the high heterogeneity of secondary outcomes and measurements related to the functional state, only stiffness was included in the quantitative analysis. QoL was measured in two articles. Yip and Tam (2008) investigated the effect of ginger and orange EOs on QoL. The results showed that EO therapy was not effective to improve QoL [29]. Pehlivan and Karadakovan (2019) concluded that aromatherapy massage improves QoL [27].

### 2.4. Quantitative Synthesis of Results

#### 2.4.1. Primary Outcome

For the analysis of pain intensity, seven articles were considered [21,24,25,26,27,28,29], with 577 patients involved in the trials. To avoid unnecessarily introduced bias, only the results of the EO therapy groups and Placebo groups were considered in the quantitative analyses of pain intensity.

Calculated mean differences (MDs), together with within-group I^2^ statistics and confidence intervals (CIs), are shown in Figure 2. Subgroups were created according to the measurement time points of the trials (i.e., immediately after the intervention or one week or four weeks after the intervention). The overall test of moderators was significant (QM = 9.98, df = 3, *p*-value = 0.0465), indicating that the time-points had an overall effect on the outcomes. The test of residual heterogeneity of the overall model was not significant (QE = 12.24, df = 9, *p* = 0.2). Model results indicate that the application of EOs was beneficial at all time points compared to placebo treatments, with significant results on week zero (i.e., immediately after the application) and week four.

##### Pain Intensity Measured Immediately after the Intervention (Subgroup Analysis)

Four trials [21,24,25,26] were included in the analysis. The MD of the change between the two groups indicates that topical EOs decreased the Visual Analogue Scale (VAS) scores significantly better than the Placebo group (MD of pain intensity = −0.87 (95% CI, −1.73 to −0.02; I^2^ = 61%; *p* = 0.014)). The difference is statistically significant between the EO group and the Placebo group.

##### Pain Intensity Measured One Week after the Intervention (Subgroup Analysis)

The results of four trials [25,27,28,29] were included for the one-week-after-intervention subgroup. Our results indicate a non-significant slight effect of EOs one week after the intervention (MD of pain intensity = −0.58 (95% CI, −1.25 to 0.10; I^2^ = 40.3%; *p* = 0.077)).

##### Pain Intensity Measured Four Weeks after the Intervention (Subgroup Analysis)

This analysis was performed on four trials [24,25,27,29]. Baseline data and data measured four weeks after the intervention were used to calculate MD between the two groups. The difference is statistically significant between the two groups (MD of pain intensity = −0.52 (95% CI, −0.96 to −0.08; I^2^ = 59.3%; *p* = 0.049)).

#### 2.4.2. Secondary Outcomes

##### Stiffness

For the analysis of stiffness, three articles were considered [27,29,31] with 124 patients involved in the trials. In the rainforest plot (Figure 3), changes in stiffness are shown one week after the intervention.

The result (MD = −0.77 (95% CI, −1.57 to 0.04; I^2^ = 68.57%; CI: 6%−96%; τ^2^ = 0.3312; *p* = 0.061)) indicates a slight improvement in the functional state of the MSD compared to no intervention. The result is nearly significant.

### 2.5. Risk of Bias Assessment and GRADE Assessment

Risk of bias assessment was performed, and all studies were evaluated to have “high risk of bias” or “some concerns”. A short summary of the performed assessment is presented in Figure 4 (intention-to-treat) and in Figure 5 (per protocol), and more details can be found in the Appendix A.

The GRADE (Grading of Recommendations, Assessment, Development and Evaluations) assessment was performed, and the overall certainty of evidence is very low in the case of both outcomes. The reasons for this may be the lack of blinding and the heterogeneity (see Appendix A).

### 2.6. Publication Bias

We used Egger’s test and the sunset funnel plot (see Appendix A) to assess potential publication bias of the meta-analysis of the primary outcome. Egger’s test was fitted by adding the SE as a moderator to the model. We found no evidence of publication bias (QM = 0.001, *p* = 0.99).

## 3. Discussion

Based on qualitative and quantitative analysis, we can conclude that EO therapy has a beneficial effect on pain intensity in MSD, and the most favorable effect was observed immediately after their usage compared to placebo. The treatment has a modest favorable effect on pain in MSDs one week and four weeks after the intervention. This seeming contradiction in results is presumably due to sample size issues, as the mean value of the effect is similar in week one compared to week four, and the *p*-value is also near significant (see also Appendix A). Nonetheless, the decrease in effect compared to week zero (i.e., immediately after the application) is apparent. The reduction of about 1 VAS score means about 10% difference in pain intensity which is a non-negligible effect. For stiffness, the results are noteworthy, albeit only marginally significant. All three involved RCTs point in the direction of the same effect and, considering that our applied methodology of conservatively estimating change standard deviations (SDs) results in a highly robust approach, we are confident that involving further analyses will yield statistically significant results.

There is a previous meta-analysis in the literature [32], in which the pain-relieving effect of aromatherapy was evaluated in all types of pain (e.g., postoperative pain, menstrual pain, knee pain). Lakhan et al. concluded that aromatherapy as an add-on treatment is effective in reducing pain.

It is known that massage therapy alone could be beneficial in MSDs via a multimodal mechanism [33,34]. Our summarized data suggest that the pain-relieving effect is more pronounced when massage is combined with an EO-containing product. The choice of EOs was based on scientific data or on traditional uses for the studies. Potential pain-relieving mechanism of EOs or EO constituents of the included clinical trials are discussed in Table A1 in the Appendix B. To reveal the differences between the effects of different EOs, more studies are needed in the future, but the tendency is obvious. EOs have beneficial effect on MSD pain and stiffness compared to placebo. Aside from EOs, other natural products may be used in the treatment of various MSDs [35,36,37,38]. However, identifying compounds with promising bioactivities is only the first step toward using them in evidence-based therapy.

### 3.1. Strengths and Limitation

Regarding the strengths of this work, we followed our protocol registered in PROSPERO [39]. Rigorous methodology was applied, and we included only RCTs in the meta-analysis. We investigated the time-dependency of the effect of EOs.

Limitations of this work are as follows: a low number of trials, involving few patients, were available in the literature, and the low-quality studies that were characterized by high risk of bias. The definition of “randomization process” differed among the studies; on some occasions it was missing. Blinding was problematic in all studies because hiding the smell of EOs was not entirely possible, and it might influence the staff and the patients. High heterogeneity was identified. The use of different EOs in the studies could explain the heterogeneity. MSDs include several conditions; consequently, the EOs were applied in different areas of the body. Moreover, the length of interventions and the follow-up periods were different.

### 3.2. Implication for Practice and Research

The main conclusion of the meta-analysis is that we were able to show the positive effect of EOs on symptoms of MSDs. No interactions were reported with the conventional therapy during the studies and, in clinical practice, the dose of painkillers might be decreased due to the pain-relieving effect of EOs. Based on the statistical analysis, repeated application of EOs is recommended at least within a week because the effect decreases after a week. It is safe, cost-effective and easily accessed by the public.

### 3.3. Recommendation for Future Trials Investigating the Effect of Topical EO on MSDs

Further high-quality RCTs with more homogeneous study designs are necessary to support the findings of this meta-analysis and to answer further questions. The most important questions concern which EOs or EO constituents have the most beneficial effect on reducing pain and stiffness and which type of MSDs can be most effectively treated with EOs. MSDs are long-term conditions; therefore, the length of the intervention and the follow-up periods should be determined carefully. Improving the methodological quality and reducing heterogeneity are important tasks in further trials. It would be advisable to devise uniform inclusion and exclusion criteria for each disorder (e.g., severity of the disease should be considered), improve blinding and provide comparable results, i.e., to reach a consensus on measurement tools intended to be used.

## 4. Methods

### 4.1. Objectives and Protocol

We report our systematic review and meta-analysis based on the recommendation of the PRISMA 2020 guideline (PRISMA checklist can be found in Appendix A) [40], and we followed the Cochrane Handbook [41]. The protocol of this systematic review and meta-analysis was registered on PROSPERO (registration number CRD42021282201) [39].

### 4.2. Information Sources and Search Strategy

Our systematic search was conducted in five different databases on 17th November, 2021. Web of Science, EMBASE, PubMed, Central Cochrane Library and Scopus were searched with the following search key: (essential oil OR aromatherapy) AND (musculoskeletal disease OR muscle OR bone OR joint) AND (topical OR cutaneous OR external OR dermal OR massage). No filters were applied.

### 4.3. Participants, Interventions, Comparisons and Outcomes (PICO)

The following PICO framework was applied to select the relevant clinical trials. Participants: adults with MSDs; Intervention: EOs applied by massage or EOs applied without massage; Comparisons: placebo product (with or without massage), or no intervention; Outcomes: pain intensity (primary outcome), quality of life (QoL) and functional state (secondary outcomes).

### 4.4. Eligibility Criteria

Only RCTs that met the established PICO were considered.

### 4.5. Exclusion Criteria

Articles were excluded based on the following criteria: animal studies; EOs administered by inhalation; no available full texts; patients studied were suffering from acute pain (trauma, injuries); patients studied were suffering from pain associated with diabetes or dysmenorrhea; the use of inappropriate placebos.

### 4.6. Selection Process

After duplicates were removed by using EndNote X9 (Clarivate Analytics, Philadelphia, PA, USA), the selection process was continued by two independent review authors (EB and EEG). The articles were selected based on title, abstract and full texts and in accordance with the predetermined inclusion and exclusion criteria. Inter-rater reliability was assessed by the calculation of Cohen’s kappa. Results of Cohen’s kappa determination showed a strong consensus degree. Disagreements were resolved by a third author (FD).

### 4.7. Data Collection Process

Data were extracted by EB and PF. Disagreements were resolved by a third reviewer (FD). Data extraction was carried out by either taking published values or, in the case of one trial [28], using a web-plot digitizer for plot reverse engineering. The studies reported the results according to different time points set; however, only the clinically relevant time points were considered, and the relating results were extracted as temporal thresholds: week zero (i.e., immediately after intervention) and one and four weeks after the intervention. Pain intensity was recorded by two scales: the VAS is a 0–10-point scale, and the Western Ontario and McMaster Universities Osteoarthritis Index (WOMAC) is a 20-point scale. Stiffness level was assessed also by two scales, i.e., by VAS (0–10) and WOMAC questionnaires (0–8). The results measured by the scales were converted to make them statistically comparable.

### 4.8. Deviation from Protocol

There were not enough data in the literature to perform the quantitative evaluation of QoL (secondary outcome). According to the inclusion criteria, studies performed on adult patients should have been selected into the meta-analysis. In the trial of Kong et al. (2012), the inclusion criterion of age was 15–35 years. In that case, we considered the patients under 18 years as young adults.

### 4.9. Study Risk of Bias Assessment

The risk of bias assessment was performed independently by EB and EEG using the Cochrane risk-of-bias tool for RCTs (RoB 2) [42]. Disagreements were resolved by FD.

### 4.10. Quality of Evidence

The GRADE approach was used to evaluate the evidence of the included trials [43]. According to GRADE, certainty of evidence of RCTs can be categorized by four categories: very low, low, moderate and high. To perform the grading, online GRADEpro GDT software was used [44].

### 4.11. Synthesis Methods

Mean before/after change difference in pain intensity measured on VAS as the primary outcome was pooled using multilevel mixed effect models [45,46]. The multilevel approach was necessary as some papers reported VAS scores for multiple time periods. Pooling mean change differences necessitates the knowledge of the SD [47] of within-group difference between time points or the correlation of within-group changes; however, most studies reported neither. In these cases, we used the sum of the reported before and after treatment group SDs as a conservative [48] estimate of variability. This approach allows us to conclude that if a result is significant with the sum of group SDs, it would certainly be significant had we used the true SDs of within group changes. To calculate the I^2^ statistic, we followed Jackson’s methodology [49]. Results are presented in rainforest plots [50] where uncertainty is visualized by the height of the raindrops for each individual estimate while the width of the raindrop corresponds to the estimated CI. All analyses were conducted in R version 4.1 [51] using the following packages: tidyverse [52], meta [53], dmetar [54] metafor [55] and metaviz [56].

## 5. Conclusions

This systematic review and meta-analysis showed that topical EOs are effective in reducing pain and stiffness in chronic MSDs, and that they contribute well to conventional therapies. Based on our results, we suggest that topical EO therapy should be applied repeatedly to reach the most effective pain-relieving effect of EOs.

However, due to the limitations of our study (low number of trials, low-quality studies with high risk of bias), further clinical investigations are needed to establish our conclusions on efficacy, to determine the most potent EOs and to understand their mode of action.

## Figures and Tables

**Figure 1 pharmaceuticals-16-00144-f001:**
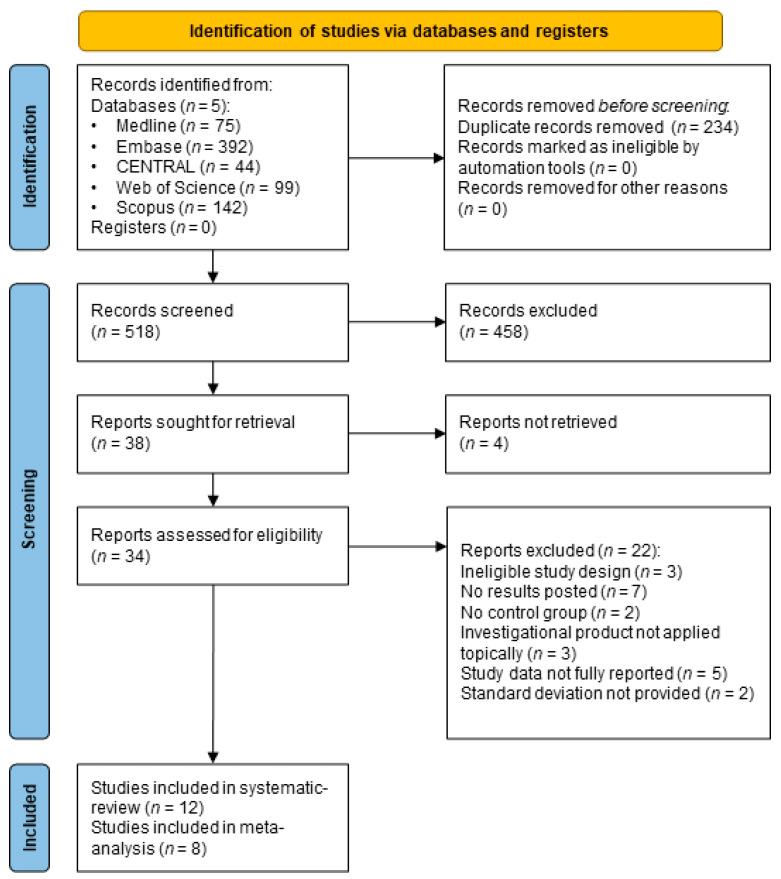
PRISMA Flowchart.

**Figure 2 pharmaceuticals-16-00144-f002:**
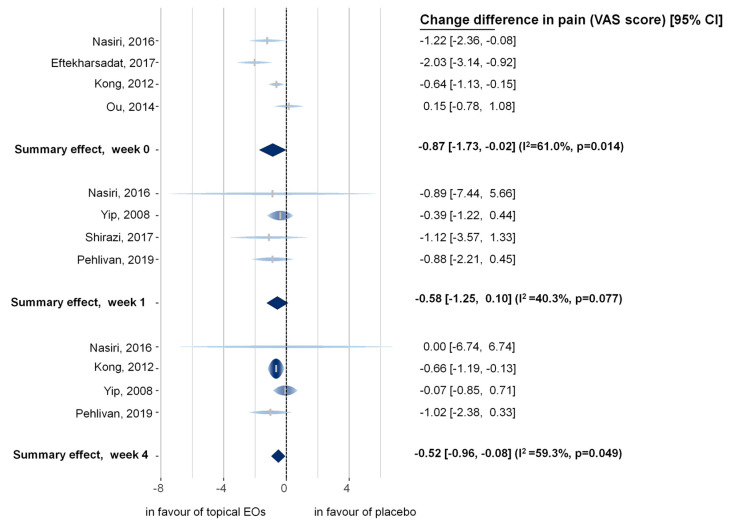
Rainforest plot of the mean difference of the changes of pain intensity. Mean difference is presented between the EO therapy group and the Placebo group at different time points. The height and color intensity of individual studies correspond to the relative importance of the study in the model. The width of the raindrop-like structures corresponds with their respective confidence intervals. CI: confidence interval; VAS: visual analogue scale; EO: essential oil; I^2^: level of heterogeneity; *p*: probability of obtaining the observed effect [21,24,25,26,27,28,29].

**Figure 3 pharmaceuticals-16-00144-f003:**
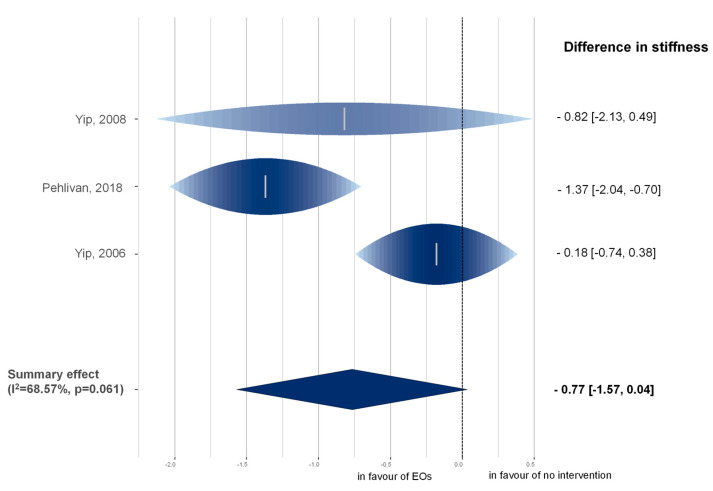
Rainforest plot of the mean difference of stiffness. Mean differences are presented between the EO therapy group and the No intervention group. The height and color intensity of individual studies correspond to the relative importance of the study in the model. The width of the raindrop-like structures corresponds with their respective confidence intervals. I^2^: level of heterogeneity; *p*: probability of obtaining the observed effect [27,29,31].

**Figure 4 pharmaceuticals-16-00144-f004:**
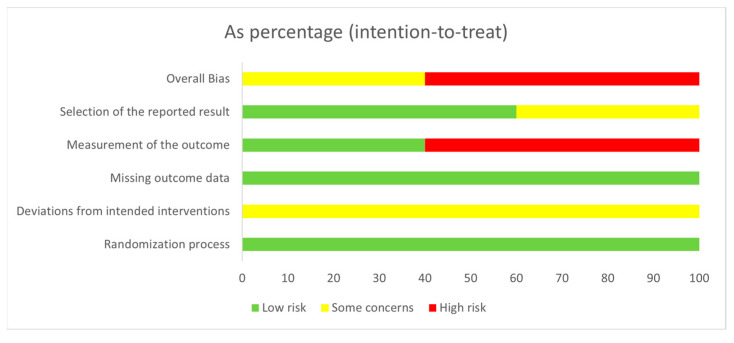
Risk of bias graphs that illustrate the proportions of studies (intention-to-treat).

**Figure 5 pharmaceuticals-16-00144-f005:**
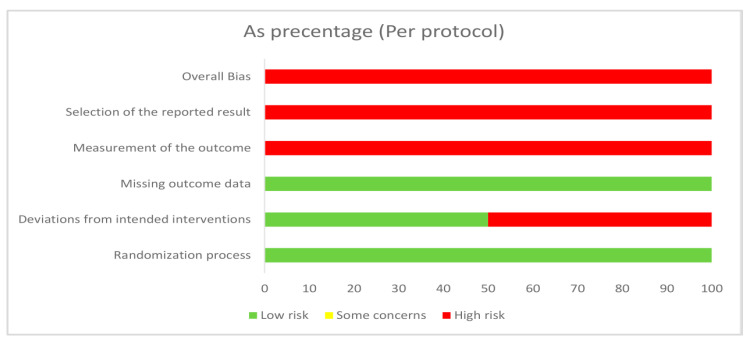
Risk of bias graphs that illustrate the proportions of studies (per protocol).

**Table 1 pharmaceuticals-16-00144-t001:** Basic characteristic of the included studies.

Study	Patients	Study Design	Country	Number of Patients	Applied Essential Oils	Intervention	Placebo	No Intervention	Outcomes
Nasiri et al., 2016 [25]	patients with knee OA	RCT	Iran	90	3% lavender oil	aromatherapy massage with lavender EO	placebo massage with sweet almond oil	no massage	pain intensity (qualitative and quantitative analysis)
Kong et al., 2012 [24]	athletes with non-specific low back pain	RCT	China	110	herbal ointment containing 20% of EOs (extracted from Dang Gui, Chuan Xiong, Xi Xin, and Rou Gui)	Chinese massage combined with herbal ointment	massage therapy with placebo ointment	n/a	pain intensity (qualitative and quantitative analysis)
Eftekharsadat et al., 2018 [21]	patients with mild to moderate CTS	RCT	Iran	48	1.5% lavender EO	night wrist orthotic and topical lavender oil ointment	night wrist orthotic and a placebo ointment	n/a	pain intensity (qualitative and quantitative analysis)
Pehlivan and Karadakovan, 2019 [27]	elderly individuals with knee osteoarthritis	RCT	Turkey	90	two EOs (2.5% ginger and 2.5% rosemary) were added to the black seed oil	aromatherapy massage	massage group (sunflower oil)	control group (no aromatherapy or massage)	pain intensity, stiffness (qualitative and quantitative analysis)
Shirazi et al., 2017 [28]	women with pregnancy-related low back pain	RCT	Iran	120	rose oil (in the carrier of almond oil)	EO applied topically	almond oil	no intervention (no EO, no massage)	pain intensity (qualitative and quantitative analysis)
Yip and Tam, 2008 [29]	moderate-to-severe knee pain among the elderly	RCT	China	59	1% ginger and 0.5% orange EO	massage with ginger and orange oil	massage intervention with olive oil only	no massage	pain intensity, stiffness (qualitative and quantitative analysis)
Ou et al., 2014 [24]	patients with neck pain	RCT	Taiwan	60	3% cream containing marjoram, black pepper, lavender and peppermint EOs	the cream was applied on the neck and upper trapezius muscles	placebo ointment	n/a	pain intensity (qualitative and quantitative analysis)
Yip and Tse, 2006 [31]	sub-acute, non-specific neck pain	RCT	China	32	3% lavender oil with olive oil	manual acupressure massage with natural aromatic lavender oil	n/a	conventional treatment	stiffness (qualitative and quantitative analysis)
Yip and Tse, 2004 [30]	non-specific low back pain	RCT	China	61	3% lavender oil with grape seed oil	acupressure massage with natural aromatic lavender oil	n/a	conventional treatment	pain intensity (qualitative analysis)
Bahr et al., 2018 [20]	hand arthritis	RCT	USA	36	mixture of EOs (main components: 16% methyl salicylate, 6% menthol, 27% beta-caryophyllene)	hand massage	coconut oil	n/a	pain intensity (qualitative analysis)
El Sayed et al., 2020 [22]	knee osteoarthritis	RCT	Egypt	60	3% lavender EO	aromatherapy massage	n/a	conventional treatment	pain intensity (qualitative analysis)

EO: essential oil; RCT: randomized controlled trial; n/a: not applicable.

## Data Availability

The datasets used in this study can be found in the full-text articles included in the systematic review and meta-analysis.

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
