# Peer review of "Efficacy of Topical Essential Oils in Musculoskeletal Disorders: Systematic Review and Meta-Analysis of Randomized Controlled Trials"

_pharmaceuticals, 2023, doi:10.3390/ph16020144_

Round 1

Reviewer 1 Report

Review the type of letters in figures and tables. 

Spelling of English language must be review.

Author Response

Response to Reviewer 1 Comments

Dear Reviewer,

thank you for reviewing paper and also for your positive comments and recommendations. Please find our answers below. We do hope that this updated manuscript meets your expectations.

Sincerely,

the authors

Point 1: Review the type of letters in figures and tables.

Response 1: Thank you for this remark. Indeed, there was a deviation in the letter types in one of the tables – this was corrected. Now we follow the MDPI template.

Point 2: Spelling of English language must be review.

Response 2: The manuscript was checked before submission by a professional language editor. However, now we checked the whole manuscript again to minimize spelling mistakes.

Reviewer 2 Report

The manuscript entitled " Efficacy of topical essential oils in musculoskeletal disorders: systematic review and meta-analysis of randomized controlled trials" deals with an interesting topic. In particular, today the use of natural substances in the treatment of human pathologies appears to be an extremely important field above all for the sustainability and reduction of the cytotoxicity of non-natural drugs.

The authors decided to report only studies conducted in humans.

However, I would have inserted a table must also mention the works in vitro and in animal models.

In addition, I would also have included a table where the targets of neuromuscular disorders were underlined and how they varied according to the type of treatment.

Finally, in the discussion I would have opened a debate on the numerous natural products that today are known to have an employment or a future use in human pathologies (doi: 10.1016/j.freeradbiomed.2022.09.017; doi: 10.1038/s41598-019-54574-4 ; doi: 10.4155/fmc.13.165; doi: 10.3390/ijms22158310)

Author Response

Response to Reviewer 2 Comments

 Dear Reviewer,

thank you for reviewing paper and also for your positive comments and recommendations. Please find our answers below. We do hope that this updated manuscript meets your expectations.

Sincerely,

the authors

Point 1: The manuscript entitled " Efficacy of topical essential oils in musculoskeletal disorders: systematic review and meta-analysis of randomized controlled trials" deals with an interesting topic. In particular, today the use of natural substances in the treatment of human pathologies appears to be an extremely important field above all for the sustainability and reduction of the cytotoxicity of non-natural drugs. The authors decided to report only studies conducted in humans. However, I would have inserted a table must also mention the works in vitro and in animal models.

Response 1: In vitro tests and animal models are important in drug development. In this meta-analysis, we have focused on topical application of essential oils. According to the literature, in the case of animal models, the most commonly used administration methods are oral or intraperitoneal methods. The following meta-analyses and reviews have been published recently:

  • Scuteri D, Hamamura K, Sakurada T, Watanabe C, Sakurada S, Morrone LA, Rombolà L, Tonin P, Bagetta G, Corasaniti MT. Efficacy of Essential Oils in Pain: A Systematic Review and Meta-Analysis of Preclinical Evidence. Front Pharmacol. 2021 Mar 1;12:640128. doi: 10.3389/fphar.2021.640128. PMID: 33732159; PMCID: PMC7957371. (Route of administration: intraperitoneal (i.p.) or subcutaneous (s.c.))
  • Assis DB, Aragão Neto HC, da Fonsêca DV, de Andrade HHN, Braga RM, Badr N, Maia MDS, Castro RD, Scotti L, Scotti MT, de Almeida RN. Antinociceptive Activity of Chemical Components of Essential Oils That Involves Docking Studies: A Review. Front Pharmacol. 2020 May 29;11:777. doi: 10.3389/fphar.2020.00777. PMID: 32547391; PMCID: PMC7272657. (Route of administration: intraocular, p.o., topical, i.p., s.c., gavage)
  • Lenardão, E. J., Savegnago, L., Jacob, R. G., Victoria, F. N., & Martinez, D. M. (2016). Antinociceptive effect of essential oils and their constituents: an update review. Journal of the Brazilian Chemical Society, 27, 435-474.(Route of administration: oral, i.p., s.c., intraplantar)
  • Ilari S, Proietti S, Russo P, Malafoglia V, Gliozzi M, Maiuolo J, Oppedisano F, Palma E, Tomino C, Fini M, Raffaeli W, Mollace V, Bonassi S, Muscoli C. A Systematic Review and Meta-Analysis on the Role of Nutraceuticals in the Management of Neuropathic Pain in In Vivo Studies. Antioxidants (Basel). 2022 Nov 28;11(12):2361. doi: 10.3390/antiox11122361. PMID: 36552569; PMCID: PMC9774415. (Route of administration: s.c. injection of bergamot EO)
  • de Cássia da Silveira E Sá, R., Lima, T. C., da Nóbrega, F. R., de Brito, A. E. M., & de Sousa, D. P. (2017). Analgesic-Like Activity of Essential Oil Constituents: An Update. International journal of molecular sciences, 18(12), 2392. https://doi.org/10.3390/ijms18122392

The manuscript has been amended by briefly mentioning the results of animal experiments.

Point 2: In addition, I would also have included a table where the targets of neuromuscular disorders were underlined and how they varied according to the type of treatment.

Response 2: Thank you for this remark. We agree that this would be an important and interesting issue, however, it is out of the scope of our meta-analysis. The potential analgesic effect of the EOs included in our meta-analysis are summarized in Table A1.

Point 3: Finally, in the discussion I would have opened a debate on the numerous natural products that today are known to have an employment or a future use in human pathologies (doi: 10.1016/j.freeradbiomed.2022.09.017; doi: 10.1038/s41598-019-54574-4 ; doi: 10.4155/fmc.13.165; doi: 10.3390/ijms22158310)

Response 3: Thank you for pointing out this aspect. We have amended and completed the manuscript according to your remark.

Reviewer 3 Report

The manuscript is very well written, the data were collected with the rigor pertinent to the process and it contributed significantly to the expansion of knowledge in the research area. However, authors need to revise some spelling details in the text for the final version of the manuscript.

Author Response

Response to Reviewer 3 Comments

Dear Reviewer,

thank you for reviewing paper and also for your positive comments and recommendations. Please find our answers below. We do hope that this updated manuscript meets your expectations.

Sincerely,

the authors

Point 1: The manuscript is very well written, the data were collected with the rigor pertinent to the process and it contributed significantly to the expansion of knowledge in the research area. However, authors need to revise some spelling details in the text for the final version of the manuscript.

Response 1: The manuscript was checked before submission by a professional language editor. However, now we checked the whole manuscript again to minimize spelling mistakes.

Reviewer 4 Report

The conclusion section can have more indication of limitations, although the authors described it in the discussion section. 

Author Response

Response to Reviewer 4 Comments

Dear Reviewer,

thank you for reviewing paper and also for your positive comments and recommendations. Please find our answers below. We do hope that this updated manuscript meets your expectations.

Sincerely,

the authors

Point 1: The conclusion section can have more indication of limitations, although the authors described it in the discussion section. 

Response 1: We agree that our conclusion should be taken with caution, so we have revised it in response to your comment.

Round 2

Reviewer 2 Report

Dear,

the manuscript can be accepted in the present form.

 The changes I requested have been made.